# The Diverse Indigenous Bacterial Community in the Rudna Mine Does Not Cause Dissolution of Copper from Kupferschiefer in Oxic Conditions

Malin Bomberg [1,*], Hanna Miettinen [1] and Päivi Kinnunen [2]

1 VTT Technical Research Centre of Finland Ltd., Tietotie 2, P.O. Box 1000, 02150 Espoo, Finland; hanna.miettinen@vtt.fi
2 VTT Technical Research Centre of Finland Ltd., Visiokatu 4, P.O. Box 1300, 33101 Tampere, Finland; paivi.kinnunen@vtt.fi
* Correspondence: malin.bomberg@vtt.fi

**Abstract:** Blasting and fracking of rock in mines exposes fresh rock surfaces to the local water and microbial communities. This may lead to leaching of metals from the rock by chemical or biological means and can cause acidification of the water system in the mine, i.e., acid rock drainage (ARD). Failure to prevent leakage of metal contaminated mine water may be harmful for the environment, especially to the local groundwater. In the Rudna mine, Poland, an in situ bioleaching pilot test at approximately 1 km depth was performed in the H2020 BIOMOre project (Grant Agreement #642456). After the leaching stage, different methods for irreversible inhibition of acidophilic iron oxidizing microorganisms used for reoxidation of reduced iron in the leaching solution were tested and were shown to be effective. However, the potential of the natural mine water microbial communities to cause leaching of copper or acidification of the mine waters has not been tested. In this study, we set up a microcosm experiment simulating the exposure of freshly fractionated Kupferschiefer sandstone or black schist to two different chloride-rich water types in the Rudna mine. The pH of the microcosms water was measured over time. At the end of an 18-week incubation, the bacterial community was examined by high throughput sequencing and qPCR, and the presence of copper tolerant heterotrophic bacteria was tested by cultivation. The dissolution of copper into the chloride rich microcosm water was measured. The pH in the microcosms did not decrease over the time of incubation. The sandstone increased the number of bacteria in the microcosms with one or over two orders of magnitude compared to the original water. The bacterial communities in the two tested mine waters were diverse and similar despite the difference in salinity. The bacterial diversity was high but changed in the less saline water during the incubation. There was a high content of sulphate reducing bacteria in the original mine waters and in the microcosms, and their number increased during the incubation. No acidophilic iron oxidizers were detected, but in the microcosms containing the less saline water low numbers of Cu tolerant bacteria were detected. Copper to a concentration of up to 939 mg L$^{-1}$ was leached from the rock also in the microbe-free negative controls, which was up to 2.4 times that leached in the biotic microcosms, indicating that the leaching was also abiotic, not only caused by bacteria.

**Keywords:** in place biomining; deep biosphere; copper ore; lithotrophic bacteria; mine microorganisms

## 1. Introduction

In place, or, in situ biomining of deeply buried ore is an attractive alternative to conventional mining by excavation, because the ore may be reached by drilling wells from the ground surface and the metals can be leached in place and pumped to the surface for concentration [1,2]. The in situ biomining has lower impact on the surrounding landscape and a smaller physical footprint compared to conventional mining that needs area for depositing waste rock and processing of mining fluids etc. in situ mining uses acidic ferric

iron solution, which has the capacity to oxidize minerals thus releasing metals from the ore into the leaching solution, which can be pumped to a processing facility [2]. The process reduces the ferric iron to ferrous form, which must be re-oxidized in order to be reused in the leaching process. In biomining this is achieved by using acidophilic iron oxidizing microorganisms, which render the lixiviant solution reusable to be pumped back into the in situ rock reactor, e.g., [3].

There are factors, however, that must be considered for in situ biomining, e.g., [4]. The microorganisms escaping the ferric iron generating bioreactor (FIGB) with the lixiviant must be inactivated after the leaching operations cease. Unwanted acid rock drainage (ARD), and the leakage of the acidic, metal rich lixiviant from the rock reactor to the environment must be prevented as to not harm the environment [5,6].

Acidophilic iron and sulphur oxidizing microorganisms are frequently found in oligotrophic, acidic environments with high concentrations of dissolved iron and sulphur [7–9]. Such environments may be found in and around mines, where sulphidic ore has been exposed to oxygen and water and a natural dissolution of the rock leads to the release of sulphuric acid, which further dissolves the rock resulting in acid rock drainage. This environment supports the growth of acidophilic iron and sulphur oxidizing microorganisms, which by oxidizing the released ferrous iron to ferric iron and reduced sulphur species to sulphuric acid continue the cycle of naturally occurring bioleaching, which releases toxic metals to the environment [10]. Uncontrolled ARD and metal dissolution have been shown to influence water bodies at several kilometers from the actual leaching site, having serious effects on the natural flora and fauna in the area [11,12].

In the H2020 BIOMOre project (https://cordis.europa.eu/project/id/642456, accessed on 14 March 2022, Grant Agreement #642456), in situ bioleaching of copper in deep subsurface deposits in the Rudna Mine, Poland, was tested. The in situ experiment consisted of the construction of an in situ rock reactor at ca. 1 km depth in the Rudna mine. The rock reactor was situated in the Kupferschiefer, which is a few meters thick layer of bituminous marl originating from ancient seabeds and has layers of dolomite, sandstone and black shale. Kupferschiefer is found over large areas of Northern and Central Europe where it is mined for e.g., silver and copper. This formation has been described in detail by Vaughan et al. [13] and the petrographic composition of the sandstone and black shale are presented in Table 1. The rock reactor with a total volume of ca. 100 m$^3$ (10 m $\times$ 5 m $\times$ 2 m, length $\times$ width $\times$ height) was constructed by fracking the black shale after which the rock reactor was rinsed with tap water to remove blasting residue and fine-grained rock material and acid washed with concentrated sulphuric acid to remove the carbonates from the rock before leaching. Leaching of copper was performed by pumping acidic ferric iron solution into the rock reactor. The solution was re-oxidized for re-use in the ferric iron oxidizing bioreactor and pumped back into the rock reactor. At the end of the leaching, the rock reactor was rinsed with concentrated sodium carbonate solution in order to stop the leaching process, rise the pH in the rock reactor and inactivate the possible remaining acidophilic iron and sulphur oxidizing bacteria left from the leaching solution.

Several laboratory tests have been performed in order to test the efficiency of irreversible inhibition of the acidophilic iron and sulphur oxidizers escaping from the bioreactor. Ballerstedt et al. [14] tested the effect of different compounds on the inhibition of the FIGB community and found that e.g., 0.4 mM formate, 300 mM chloride or 100 mM nitrate, different alcohols, SDS and benzoic acid were effective, whereas 0.6 mM acetic acid or 200 mM methanol were not. Bomberg et al. [15] showed in a microcosm experiment that local chloride-rich mine water, containing 1.8 M Cl$^-$ was also efficiently inhibiting the FIGB community. Nevertheless, Zhang et al. [16] showed in a laboratory experiment with a FIGB simulating the one used in the in situ experiment in Rudna, that the mesophilic, acidophilic iron oxidizers from the FIGB were able to continue growth and iron reduction at pressure as high as 36 MPa, although without inhibiting compounds, such as chloride. They also enriched a thermophilic acidophilic consortium consisting mostly of *Thermoplasma* species

from the local Kupferschiefer from the Rudna mine, thus showing that such microbial communities inhabit the ore and that they also remain live and active at 36 MPa pressure.

Previous studies have shown a surprisingly rich community of culturable heterotrophic copper resistant bacteria in the Lubin mine, which like Rudna excavates the Kupferschiefer [17,18]. Copper is generally toxic to microorganisms, e.g., [19,20]. Thus, Cu tolerant or resistant microorganisms may have a significant impact on the possible ARD and may present a competitive edge against microorganisms with less tolerance towards Cu.

In this study, we simulated the situation where local saline water in the mine comes into contact with the newly fracked rock, here Kupfershiefer sandstone and black shale, using two different Rudna mine water types in a microcosm setup. We investigated the aptitude of the indigenous microorganisms inhabiting the saline water to leach copper from the rock, affect the pH of the water and cause ARD as well as the effect of the rock material on the size and composition of the microbial communities in the microcosms using qPCR and high-throughput amplicon sequencing. We also tested the water for the presence of acidophilic iron oxidizing microorganisms and copper tolerant microorganisms by cultivation.

**Table 1.** Physicochemical parameters of the mine waters and the petrographic composition of the rock types from [21].

| Parameter | Water 1 | Water 2 | Unit |
|---|---|---|---|
| pH | $6.3 \pm 0.5$ | $7.4 \pm 0.6$ | |
| Dissolved substances | $119.5 \pm 13.1$ | $40\,825 \pm 4\,491$ | $mg\,L^{-1}$ |
| Volatile substances | 5.4 | 1.8 | $g\,L^{-1}$ |
| Mineral substances | 114.1 | 39 | $g\,L^{-1}$ |
| $Cl^-$ | $64.5 \pm 6.5$ | $19.1 \pm 1.9$ | $g\,L^{-1}$ |
| $SO_4^{2-}$ | $3.2 \pm 0.6$ | $2.9 \pm 0.6$ | $g\,L^{-1}$ |
| Fe | <0.05 | <0.05 | $mg\,L^{-1}$ |
| Ca | $3.3 \pm 0.4$ | $1.8 \pm 0.2$ | $g\,L^{-1}$ |
| Na | 38.9 | 10.7 | $g\,L^{-1}$ |
| Cu | 0.3 | 0.04 | $mg\,L^{-1}$ |
| As | <0.005 | $0.005 \pm 0.001$ | $mg\,L^{-1}$ |
| Zn | 22.0 | 0.9 | $mg\,L^{-1}$ |
| Ni | 0.1 | <0.030 | $mg\,L^{-1}$ |
| Hg | <0.5 | <0.5 | $mg\,L^{-1}$ |
| Pb | 10.0 | <0.070 | $mg\,L^{-1}$ |
| V | <0.01 | <0.01 | $mg\,L^{-1}$ |
| | **Sandstone** | **Black shale** | |
| Quartz | 72 | 5 | % |
| Clay minerals | 15 | 39 | % |
| Carbonates | 8 | 42 | % |
| Sulphuros minerals | 2 | - | % |
| Organic compounds | Trace amounts | 6 | % |
| Copper sulphides | 3 | 8 | % |

## 2. Materials and Methods

### 2.1. Sampling and Preparations

Water from two water collection ponds in Rudna mine, Water 1 and Water 2, was collected on 20 April 2016 in sterile 1 L Nalgene bottles, kept cooled at +4 °C and shipped on coolers to Finland. Upon arrival, duplicate 1000 mL samples were filtered on Sterivex filter units, pore size 0.02 μm (Merck Millipore, Burlington, MA, USA), in order to collect the biomass. The filter units were frozen at −20 °C until DNA extraction. The remaining water was stored at +4 °C until construction of the microcosms. The pond surfaces were constantly exposed to the ambient oxygen-containing atmosphere in the mine.

Sandstone and black shale of the Rudna mine Kupferschiefer were roughly crushed with a sterile sledgehammer in sterile plastic bags into approximately 0.5–4 cm diameter

pieces, which were rinsed with sterile MilliQ water to remove the finest particles. The rock was allowed to dry for 3 days at 37 °C before the use in the microcosms.

### 2.2. Microcosm Experiment

Microcosm experiments were set up in autoclaved 250 mL borosilicate bottles (Schott, Mainz, Germany) containing 100 g sandstone (SS) or black shale (BS). Thereafter, 100 mL of Water 1 or Water 2 was added to the microcosms and the bottles were loosely sealed with screw caps in order to allow for some passive ventilation but preventing contamination and excessive evaporation. Microcosms containing no rock (NR) material were prepared as water controls, and in addition, abiotic control microcosms with mine water with or without rock material treated with glutaraldehyde (GDH) to a final concentration of 0.1% *v/v* were prepared. The microcosms were incubated for a total of 18 weeks at 30 °C. All biotic microcosms and the abiotic microcosms with Water 2 were prepared in duplicate, whereas only one microcosm was prepared for the abiotic microcosms containing Water 1, due to shortage of water sample.

Aliquot samples of 5 mL were collected from all bottles after 2.5, 7.5 and 18 weeks of incubation for measuring the pH development over time with a pH probe (Denver Instruments, Bohemia, NY, USA) at room temperature.

At the end of the microcosm incubation, 10 mL aliquot samples were retrieved from each microcosm bottle and collected on Sterivex filters for DNA extraction.

### 2.3. Cultivation Experiments

The presence of culturable copper resistant microorganisms in the microcosms was examined by plate counts. R2A media (Merck, Kenilworth, NJ, USA) was prepared containing 10% or 3% NaCl to complement for the salinity of the Water 1 and Water 2, respectively. The agar plates were further amended with $CuSO_4 \cdot 5H_2O$ to a final concentration of 0 mM, 0.5 mM and 100 mM. Aliquots of 100 μL of undiluted, 100-fold and 10,000-fold diluted sample water were spread on duplicate agar plates from all microcosms. The plates were incubated at 30 °C.

The presence and activity of acidophilic, iron oxidizing microorganisms in the microcosms at the end of incubation was tested by inoculating 900 μL acidic, iron oxidizer medium [22] with 100 μL of water from each microcosm. The medium consisted of 3.75 g $L^{-1}$ $(NH_4)_2SO_4$, 1.875 g $L^{-1}$ $Na_2SO_4 \cdot 10H_2O$, 0.625 g $L^{-1}$ $MgSO_4 \cdot 7H_2O$, 0.125 g $L^{-1}$ KCl, 0.0625 g $L^{-1}$ $K_2HPO_4$, and 0.0175 g $L^{-1}$ $Ca(NO_3)_2 \cdot 4H_2O$, supplemented with 1% (*v/v*) trace element solution consisting of 1.375 g $L^{-1}$ $FeCl_3 \cdot 6H_2O$, 0.319 g $L^{-1}$ $MnSO_4 \cdot 4H_2O$, 0.25 g $L^{-1}$ $H_3BO_3$, 0.1125 g $L^{-1}$ $Na_2SeO_4$, 0.1125 g $L^{-1}$ $ZnSO_4 \cdot 7H_2O$, 0.1 g $L^{-1}$ $Na_2MoO_4 \cdot 2H_2O$, 0.075 g $L^{-1}$ $CoCl_2 \cdot 6H_2O$ and 0.0625 g $L^{-1}$ $CuSO_4 \cdot 5H_2O$. Ferrous iron, 5.6 g $L^{-1}$, was added to the medium as $FeSO_4 \cdot 7H_2O$. Duplicate cultivations were prepared for each microcosm. The cultures were incubated at 30 °C for two weeks whereafter they were visually examined for colour change of the medium from clear to rust colour, indicating aptitude for iron oxidation activity in the sample. The colour of the cultures was compared to a positive control consisting mostly of *Leptospirillum*, which was the effluent from a laboratory scale ferric iron generating bioreactor (FIGB) [15]. The positive control cultures were prepared in the same way as the cultures from the microcosms, and the uninoculated culture medium incubated together with the samples served as negative control.

### 2.4. Measurement of Copper Concentration

The concentration of copper in the microcosms water at the end of the incubation was measured using LCK 329 Copper cuvette test (Hach Company, Loveland, CO, USA). Sample water from the microcosms was diluted in to 1:1000 in MilliQ water before used for the Cu measurements. For each test, 2 mL 1:1000 diluted, non-filtered sample was used and the preparations and measurements proceeded according to the manufacturer's instructions.

*2.5. DNA Extraction, qPCR and Amplicon Sequencing*

DNA was extracted from the mine water and microcosm water samples using the NucleoSpin Soil DNA extraction kit (Macherey-Nagel Gmbh & Co. KG, Düren, Germany) using the SL1 lysis buffer and Enhancer solution. The thawed membranes were placed in sterile 5 mL Eppendorf tubes to which the beads of one bead tube and the recommended volumed of SL1 buffer and Enhancer solution were added. The tubes were placed horizontally on a Vortex Genie 2.0 (Scientific Industries, Inc., Bohemia, NY, USA) shaker and mixed at full speed for 5 min. The tubes were then centrifuged on an Eppendorf 5810R benchtop centrifuge at $3184 \times g$ for 5 min. Thereafter the supernatant was transferred to new 2 mL tubes provided with the extraction kit and the DNA extraction procedure continued according to the manufacturer's instructions. The DNA was eluted in 100 µL elution buffer. The DNA concentration of the extractions was measured using the Qbit (Invitrogen, Life Technologies Corporation, Eugene, OR, USA) spectrophotometer with the dsDNA HS Assay kit.

The bacterial community size was estimated using quantitative PCR (qPCR) targeting the bacterial 16S rRNA gene with primers Bact_341F/Bact_805R [23]. In addition, the abundance of sulphate reducing bacteria in the water samples was estimated by targeting the bacterial *dsr*B gene with primers 2060r and 4F [24]. For all samples, triplicate 10 µL reactions were run using the SensiFAST™ Real-Time PCR Kit (Bioline, London, UK) containing 1 µL DNA template and 1 µM of each relevant primer. Negative reagent control reactions were included in each qPCR run. The amplification results were compared to those of standard curve consisting of a 10-fold dilution series ($10^1$–$10^8$ copies/reaction) of plasmids containing the target 16S rRNA gene fraction of *Escherichia coli* (ATCC 31608) or the *dsr*B gene of *Desulfobulbus propionicus* DSM 2554, respectively. The amplification program consisted of an initial denaturation step of 15 s at 95 °C followed by 45 amplification cycles consisting of a denaturation at 95 °C for 10 s, annealing at 57 °C and an extension step of 30 s at 72 °C, with fluorescence measurement at the end of each extension. The amplification cycles were followed by a final extension for 3 min at 72 °C and melting curve analysis. For the melting curve analysis, the amplicons were first denatured at 95 °C, followed by an annealing step at 60 °C and a progressive denaturation to 95 °C with a temperature increase of 2.2 °C s$^{-1}$ and continuous fluorescence measurement. All qPCR assays were run on the LightCycler480 instrument (Roche Diagnostics International AG, Rotkreuz, Switzerland).

The bacterial communities in the two mine waters and microcosm waters incubated for 18 weeks were characterized using Iontorrent high throughput amplicon sequencing. Bacterial 16S rRNA gene amplicons were obtained using primers Bact_341F/Bact_805R [23], targeting the variable region V3–V4 region of the bacterial 16S rDNA gene. The primers were equipped with adapter sequences for the Iontorrent platform at their 5′ ends and the forward primers also had 9-nucleotides long barcodes unique for each sample. The PCR amplifications were performed in 25 µL reactions and parallel reactions were prepared for each sample, whereafter the amplifications were combined in order to diminish PCR biases. The amplification reaction mix consisted of 1x MyTaq™ Red Mix (Bioline, London, UK), 20 pmol of each primer, up to 25 µL molecular-biology-grade water (MilliporeSigma, St. Louis, MO, USA) and 2 µL of template DNA. The PCR reactions were run on an Eppendorf MasterCycler gradient thermocycler (Eppendorf, Hamburg, Germany) and the program contained an initial denaturation step at 95 °C for 3 min, an amplification program of 15 s at 95 °C, 15 s at 50 °C and 15 s at 72 °C, run for 35 cycles for bacteria and fungi and 40 for archaea, with a final elongation of 30 s at 72 °C. Correct sizes of the amplicons were verified with agarose gel electrophoresis. Amplicons were sent to Ion Torrent PGM (Thermo Fisher Scientific, Waltham, MA, USA) sequencing on a 316 chip to Bioser Oy (Oulu, Finland) where the amplicons were purified and size checked before sequencing.

*2.6. Sequence Data Analysis*

The sequences data was analysed using the mothur software package version 1.43.0 [25] using the Silva version 138 16S rRNA gene database [26,27]. The adapters and barcodes were first removed whereafter the sequence reads were trimmed and retained according to the quality requirements, i.e., minimum length of 200 bp, maximum one nucleotide difference in the primer sequence and no mismatches in the barcode sequences, homopolymer stretches of a maximum of 8 nucleotides, no ambiguous nucleotides, a quality average of 20 over 40 nucleotide window size. The sequence data was dereplicated using unique.seqs whereafter sequences were aligned using align.seqs against the Silva 138 full database that had been optimized to cover only the region flanked by the primers. The alignment was subsequently screened using screen.seqs to include only sequences covering the targeted area of the alignment, i.e., starting by the latest at position 6430 and ending at the earliest by position 14,000. The screened alignments were filtered using filter.seqs to remove gaps spanning all sequences and cut the ends of the alignment containing no nucleotides. Redundancy of the filtered sequences was again removed using unique.seqs and possible sequencing errors were removed using the pre.cluster command. Chimeric sequences were identified using the chimera.vsearch command and the chimeric sequences were removed from the data. The sequence reads were classified using the Silva 138 database after which all non-bacterial sequences were removed. A distance matrix was built on the aligned bacterial sequences using the dist.seqs command, and the sequences were clustered into OTUs sharing 97% sequence homology using the cluster command. A table of the abundance of each OTU in each sample was produced using the make.shared command and the OTUs were classified based on the sequence read classification. Finally, a biom table was constructed, which was imported into phyloseq [28] in R [29] for further analysis.

*2.7. Statistical Analysis*

The statistical difference in number of bacterial 16S rRNA gene copies and *dsr*B gene copies in the different microcosm types was tested using Tukey's pairwise comparison test in PAST v 3.0 [30]. Alpha- and betadiversity, i.e., estimated number of OTUs using the Chao1 estimator, Shannon's diversity index and principal coordinates analysis (PCoA) with the Bray Curtis dissimilatory model and pH and Cu concentration as environmental factors was performed using phyloseq in R.

**3. Results**

*3.1. pH, Cu Content, Cu Resistant Microorganisms and Iron Oxidation*

The pH in the original mine waters was $6.3 \pm 0.5$ and $7.4 \pm 0.6$ in Water 1 and Water 2, respectively (Table 1). In Water 1 the pH increased slightly to a maximum of $7.2 \pm 0.4$ in the rock containing microcosms and to $7.5 \pm 0.2$ in the no-rock microcosms (Figure 1). In the Water 2 microcosms, the pH increased to 7.8 and 7.9 in the sandstone and black shale microcosms, respectively, over the first 2.5 weeks but returned to 7.3–7.4 at the end of the incubation. The no-rock microcosms had a higher pH of ca 8.0 at the beginning of the incubation and decreased to 7.7 by the end of the incubation. The pH was in general lower in the abiotic controls.

The original mine waters contained 0.3 and 0.04 mg Cu $L^{-1}$, in Water 1 and Water 2, respectively (Table 1). The no-rock microcosm waters contained 29 and 48 mg Cu $L^{-1}$, according to the Hach LCK 329 kit, indicating an overestimation of the Cu concentration likely because the samples measured were not filtered before measuring (Table 2). However, more Cu was leached from the rock material in the Water 1 than in the Water 2 microcosms.

A low number of culturable heterotrophic Cu resistant microbial colonies grew on the CuSO$_4$ amended R2A agar plates (Table 2). Colony forming units (cfu) were more frequent in sandstone microcosms and more commonly found in Water 2 microcosms than in Water 1 microcosms (Table 2).

The cultivation-based test for detecting presence of acidophilic iron oxidizing microorganisms in the mine water microcosms was negative for the mine water samples and microcosms, but positive for the positive control.

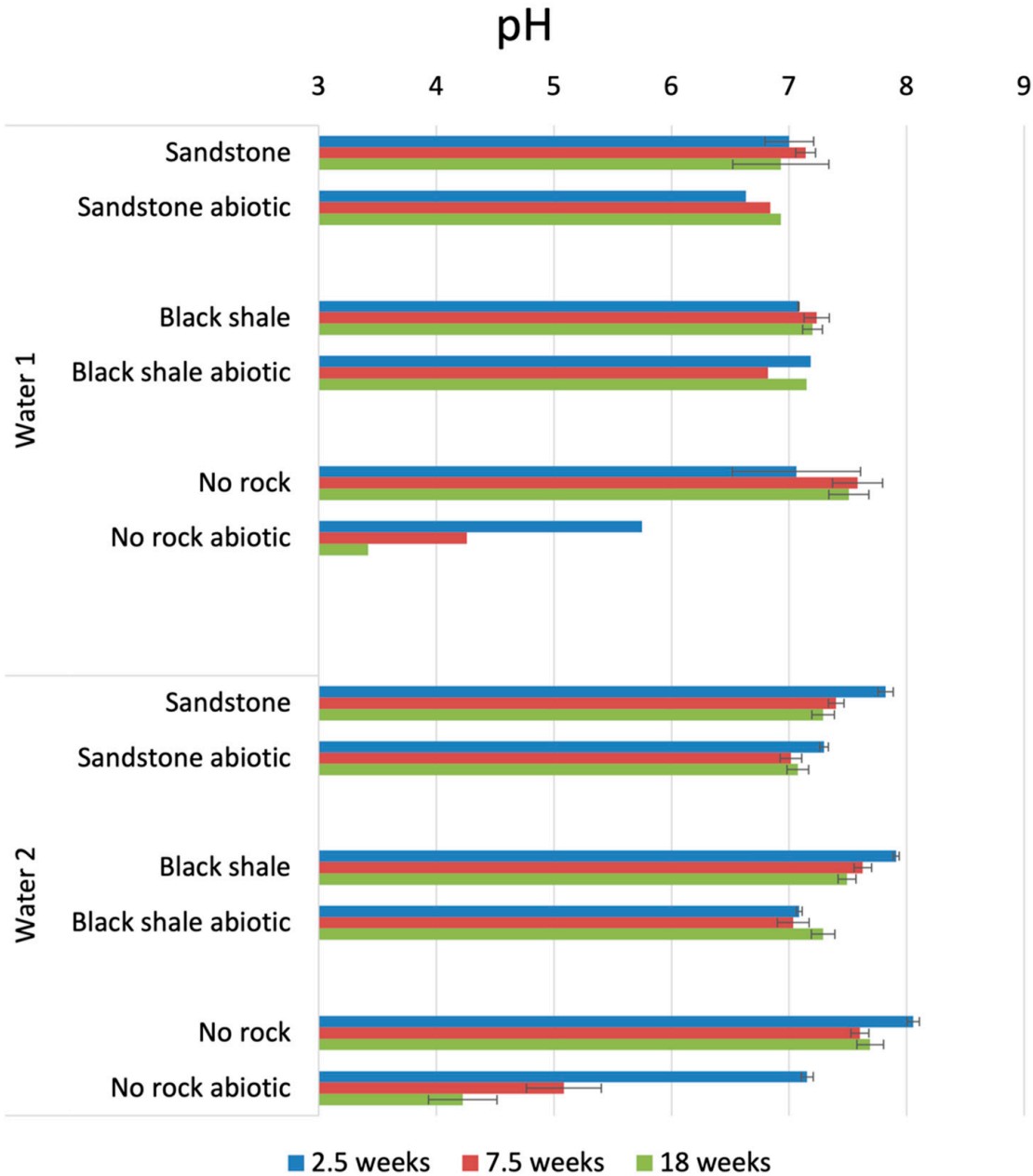

**Figure 1.** The average pH values in the water of the live and abiotic microcosms after 2.5, 7.5 and 18 weeks of incubation. Error bars indicate highest and lowest values of two microcosms. Abiotic controls for Water 1 show measurements of one microcosm each.

*3.2. Sequence Data*

Between 3009 and 21,934 sequence reads were obtained from the samples (Table 3). These were grouped into 255 to 3982 operational taxonomic units (OTUs) with 97% internal sequence homology. The Chao1 estimated number of OTUs was between 469 and 5086. Both the number of detected OTUs and Chao1 estimated number of OTUs were generally higher in the Water 1 water and microcosms compared to the Water 2 counterparts, with some exceptions (Table 3). Shannon's diversity index varied between 2.8 and 7.2. The lowest number of OTUs and Chao1 estimated OTUs as well as the lowest Shannon diver-

sity indexes were calculated for the Water 2 microcosms and one of the Water 1 no-rock microcosms (Table 3).

**Table 2.** Copper content and number of Cu tolerant culturable, heterotrophic bacteria mL$^{-1}$ microcosm water at the end of the incubation in the biotic (replicates A and B) and abiotic microcosms. Note that no rock microcosms are represented by single microcosms. bd–below detection limit ($<10^3$ mL$^{-1}$).

| Sample | Microcosm Type | Cu Concentration mg L$^{-1}$ | Cfu mL$^{-1}$ on 0.5 mM (92 mg L$^{-1}$) Cu Plates |
|---|---|---|---|
| Water 1 SS A | Sandstone | 465 | $0.5 \times 10^4$ |
| Water 1 SS B | Sandstone | 507 | bd |
| Water 1 SS abio | Sandstone abiotic | 939 | bd |
| Water 1 BS A | Black shale | 297 | bd |
| Water 1 BS B | Black shale | 146 | bd |
| Water 1 BS abio | Black shale abiotic | 231 | bd |
| Water 1 NR A | No rock | 29 | bd |
| Water 1 NR abio | No rock abiotic | 41 | bd |
| Water 2 SS A | Sandstone | 154 | $5.5 \times 10^4$ |
| Water 2 SS B | Sandstone | 159 | $4.5 \times 10^4$ |
| Water 2 SS abio | Sandstone abiotic | 347 | bd |
| Water 2 BS A | Black shale | 121 | $4.0 \times 10^4$ |
| Water 2 BS B | Black shale | 98 | bd |
| Water 2 BS abio | Black shale abiotic | 262 | bd |
| Water 2 NR A | No rock | 48 | $1.0 \times 10^4$ |
| Water 2 NR abio | No rock abiotic | 23 | bd |

**Table 3.** Sequence data information and alphadiversity.

| Sample | Original Water or Microcosm Type | Number of Sequences | Number of OTUs | Chao1 Estimated Number of OTUs | Shannon's Diversity Index |
|---|---|---|---|---|---|
| Water 1 A | Original | 8946 | 2406 | 3761 | 6.9 |
| Water 1 B | Original | 8776 | 2452 | 3775 | 6.9 |
| Water 1 SS A | Sandstone | 21,934 | 3982 | 5086 | 7.2 |
| Water 1 SS B | Sandstone | 7899 | 2324 | 3597 | 6.9 |
| Water 1 BS A | Black shale | 5840 | 1869 | 2780 | 6.8 |
| Water 1 BS B | Black shale | 3081 | 1247 | 2304 | 6.6 |
| Water 1 NR A | No rock | 3009 | 1248 | 2151 | 6.6 |
| Water 1 NR B | No rock | 5822 | 372 | 661 | 2.9 |
| Water 2 A | Original | 10,614 | 2779 | 4048 | 7.0 |
| Water 2 B | Original | 2145 | 967 | 1946 | 6.4 |
| Water 2 SS A | Sandstone | 5399 | 328 | 533 | 2.8 |
| Water 2 SS B | Sandstone | 7166 | 413 | 587 | 2.9 |
| Water 2 BS A | Black shale | 3340 | 265 | 559 | 3.4 |
| Water 2 BS B | Black shale | 4141 | 296 | 656 | 3.4 |
| Water 2 NR A | No rock | 4496 | 319 | 565 | 3.3 |
| Water 2 NR B | No rock | 4281 | 255 | 469 | 2.9 |

### 3.3. Microbial Communities

The bacterial communities in the two mine waters did not differ greatly in the main community structure (Figure 2), but the PCoA analysis (Figure 3) placed one Water 2 mine water sample, both Water 1 black shale and one of the Water 1 no-rock samples separately on the lower left corner of the plot, whereas both Water 1 mine water samples, the other Water 2 mine water sample and both Water 1 sandstone microcosm samples in the upper left corner of the plot. All Water 2 microcosm samples and one of the Water 1 no-rock samples formed a tight group on the right-hand side of the plot.

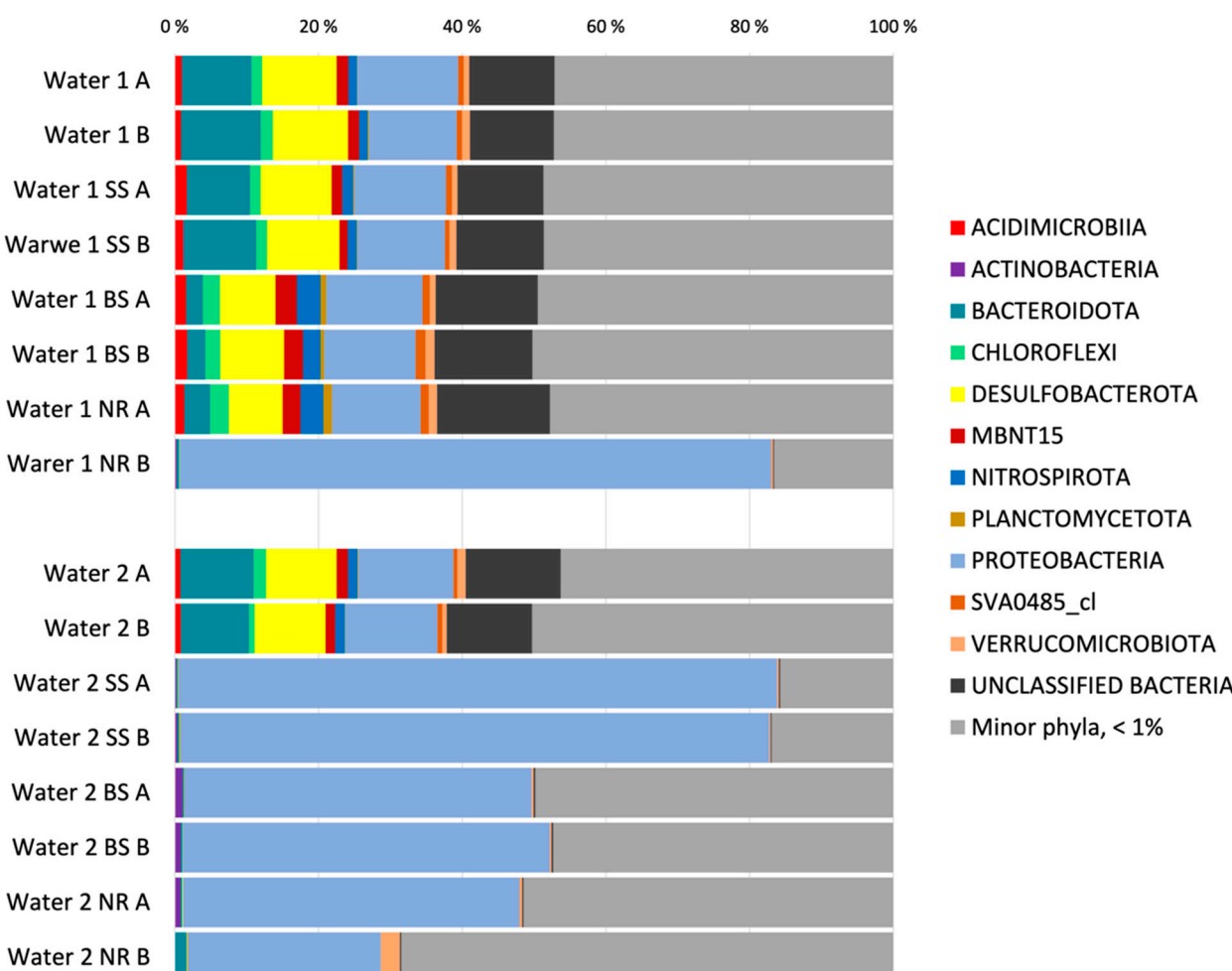

**Figure 2.** The relative abundance of bacterial phyla in the mine water and microcosm waters incubated for 18 weeks. The combined relative abundance of minor genera that each contributed with less than 1% of the bacterial communities are collected under Minor phyla, <1%. SS, BS and NR indicate microcosms containing sandstone, black shale and no rock, respectively, and A and B indicate replicate microcosms.

Altogether, 965 different bacterial genera were identified from the mine water and microcosm samples. Of these, only 46 genera were present at more than 1% relative abundance in at least one sample. Thus, between 15.7 and 68.5% of the sequence reads belonged to minority genera (Table 4). In addition, between 0.14% and 15.7% of the bacterial sequence reads remained without specific taxonomical assignments, i.e., unclassified bacteria.

The bacterial communities in the two mine waters were very similar (Figures 2 and 4. In addition to the combined great relative abundance of minority groups and unclassified bacteria, the major identifiable phyla were Bacteroidota, Desulfobacterota and Proteobacteria (Figure 2), which contributed with 9.6–14.0% of the bacterial sequence reads, each, in the original mine water. In the Water 1 microcosms, the bacterial community structure remained similar to the original mine water over the 18 weeks of incubation, with the exception of one of the no-rock replicate microcosms. In the Water 2 microcosms, the bacterial community changed radically and was dominated by Proteobacteria, contributing with 26.86–83.36% of the bacterial communities after 18 weeks of incubation. The commu-

nities in the Water 2 microcosms were similar to the community in the divergent Water 1 no-rock microcosms.

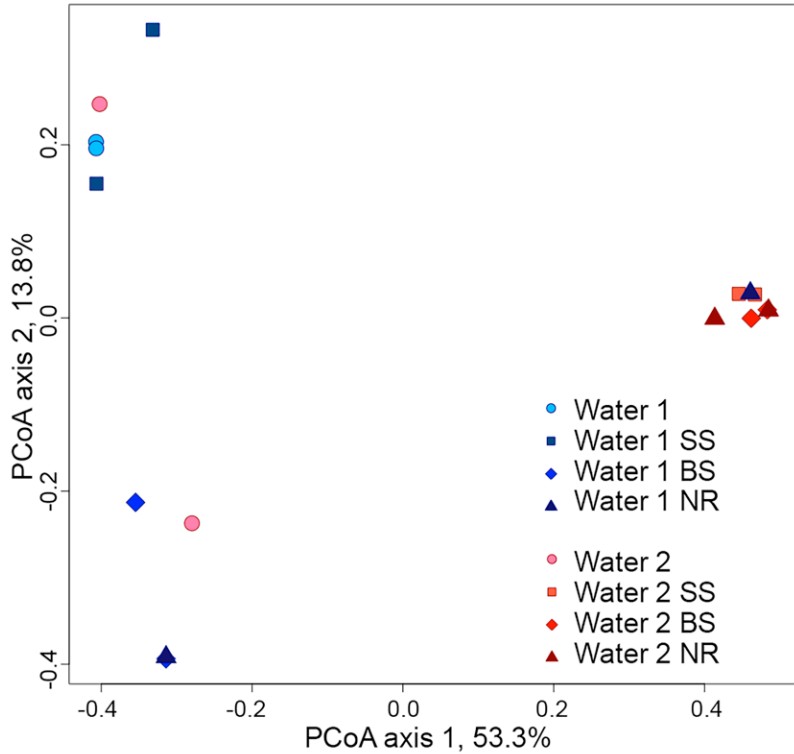

**Figure 3.** Principal coordinates analysis (PCoA) of the bacterial communities in the mine waters and 18-week microcosms based on unnormalized sequence read counts and Bray-Curtis dissimilarity model. Blue hues display Water 1 mine water and microcosms and red hues Water 2 mine water and microcosms. SS, BS and NR indicate microcosms containing sandstone, black shale and no rock, respectively. The percentage values on the axes indicate percentage of variation.

**Table 4.** Relative amounts of sequence reads of unclassified bacteria, reads belonging to genera contributing with less than 1% of the sequences in any sample or belonging to major genera. SS—sandstone, BS—black shale, NR—no rock.

| Sample | % Sequences of Unclassified Bacteria | % Sequences Belonging to Minor Genera, i.e., <1% of the Community | % Sequences Belonging to Major Genera |
|---|---|---|---|
| Water 1 A | 11.9 | 47.1 | 41.0 |
| Water 1 B | 11.7 | 47.2 | 41.1 |
| Water 1 SS A | 12.0 | 48.6 | 39.3 |
| Water 1 SS B | 12.2 | 48.6 | 39.2 |
| Water 1 BS A | 14.2 | 49.5 | 36.3 |
| Water 1 BS B | 13.7 | 50.2 | 36.1 |
| Water 1 NR A | 15.7 | 47. 8 | 36.5 |
| Water 1 NR B | 0.2 | 16.6 | 83.3 |
| Water 2 A | 13.3 | 46.2 | 40.5 |
| Water 2 B | 11.9 | 50.2 | 37.8 |
| Water 2 SS A | 0.2 | 15.7 | 84.1 |
| Water 2 SS B | 0.1 | 16.9 | 83.0 |
| Water 2 BS A | 0.2 | 49.8 | 49.9 |
| Water 2 BS B | 0.2 | 47.3 | 52.4 |
| Water 2 NR A | 0.2 | 51.4 | 48.3 |
| Water 2 NR B | 0.2 | 68.5 | 31.3 |

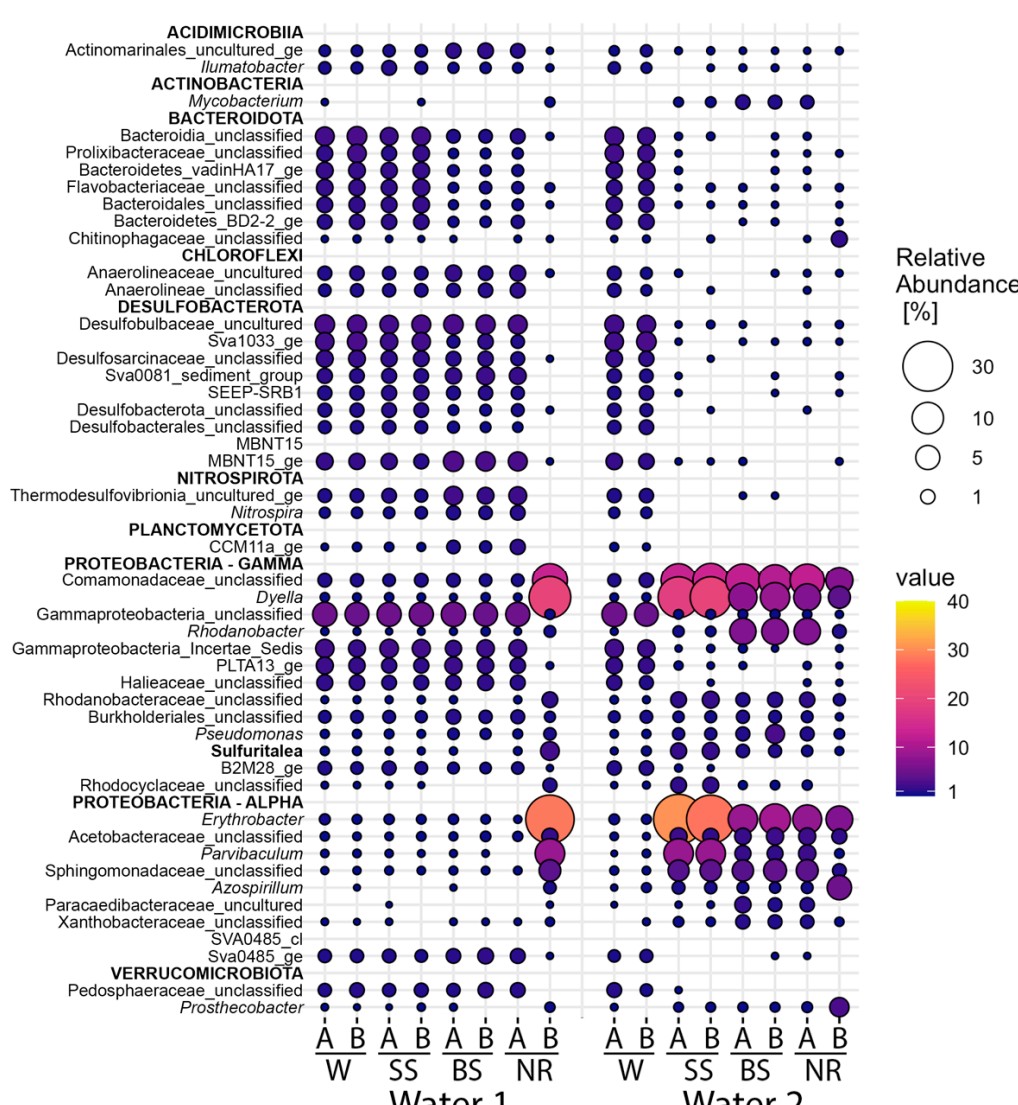

**Figure 4.** The relative abundance of the most common bacterial genera in the mine waters and microcosm water samples at the end of incubation. The colour scale and size of the dots indicate the relative abundance of each identified genus in each sample, according to the rulers on the right of the graph. The relative abundances of unclassified bacteria and the combined relative amounts of minor genera that each contributed with less than 1% of the bacterial communities in all samples are presented in Table 4. W, SS, BS and NR indicate original mine water (W), microcosms containing sandstone (SS), black shale (BS) and no rock (NR), respectively, and A and B indicate replicate microcosms.

Interestingly, an average of 11.2% and 10.7% of the bacterial sequence reads in the original Water 1 and Water 2 belonged to sulphate reducing bacteria (SRB), including the sequences affiliating with the phylum Desulfobacterota and the unclassified Thermodesulfovibrionia genus of thee Nitrospirota phylum. In the Water 1 microcosms, especially the ones containing rock, the relative abundance of SRB remained at an average of 10.5% of the total bacterial community over the time of incubation, whereas in Water 2 microcosms, the relative abundance of SRB decreased to below 0.1% (Figure 4).

### 3.4. Abundance of Microorganisms

The number of bacterial 16S rRNA gene copies in the mine water was $3.5 \times 10^5$ and $2.5 \times 10^4$ 16S rRNA gene copies $mL^{-1}$ in the Water 1 and Water 2, respectively

(Figure 5). In the microcosms with sandstone, the bacterial 16S rRNA gene numbers were $1.9 \times 10^6$ and $7.2 \times 10^6$ copies mL$^{-1}$ in Water 1 and Water 2, respectively, and in the black shale microcosms the corresponding numbers were $1.5 \times 10^5$ and $1.4 \times 10^6$ copies mL$^{-1}$, respectively. The no rock microcosms had $3.4 \times 10^4$ and $3.9 \times 10^4$ copies mL$^{-1}$ in Water 1 and Water 2, respectively. Tukey's pairwise range test (Tukey's Q) showed that the Water 2 sandstone microcosms had statistically significantly higher number of bacterial 16S rRNA gene copies than any of the other samples ($p < 0.005$).

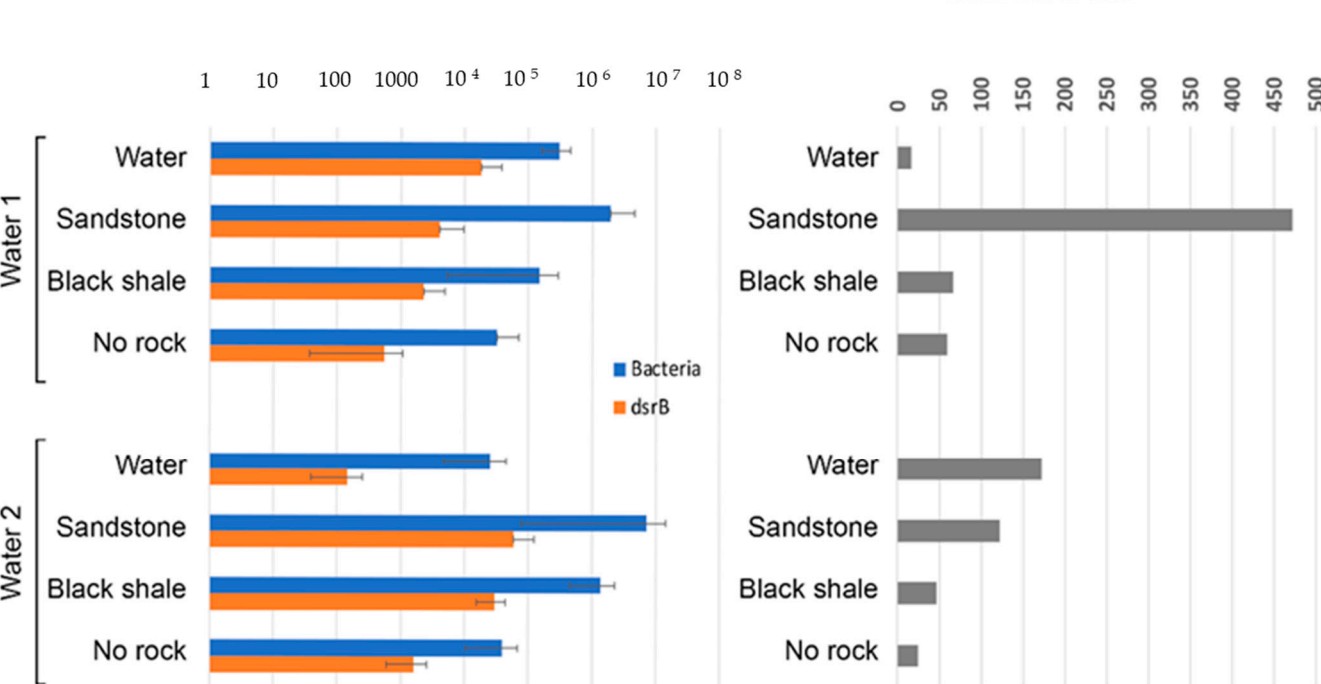

**Figure 5.** The average number of bacterial 16S rRNA gene and *dsr*B gene copies of sulphate reducers mL$^{-1}$ in the mine waters and microcosm waters (**left**) and the ratio of 16S rRNA gene copies to *dsr*B gene copies (**right**). The measurements are based on one sample from two replicate microcosms and three replicate qPCR reactions per sample at the end of the incubation. The error bars indicate standard deviation (STD).

Due to the surprisingly high relative abundance of bacterial 16S rRNA gene sequences belonging to sulphate reducing bacterial taxa, the number of *dsr*B genes in the samples were determined with *dsr*B gene targeting qPCR. *dsr*B gene copies were detected from both mine waters (Figure 5). Water 1 contained $1.8 \times 10^4$ *dsr*B gene copies mL$^{-1}$ whereas Water 2 had $1.5 \times 10^2$ *dsr*B gene copies mL$^{-1}$. In the Water 1 microcosms the *dsr*B gene content decreased from $4.1 \times 10^3$ copies mL$^{-1}$ in the sandstone microcosms to $2.3 \times 10^3$ copies in the black shale and $5.5 \times 10^2$ copies mL$^{-1}$ in the no-rock microcosm after 18 weeks of incubation. In the Water 2 microcosms the *dsr*B gene counts increased, the sandstone microcosms had $5.9 \times 10^4$ copies mL$^{-1}$, the black shale $3.0 \times 10^4$ and the no-rock $1.6 \times 10^3$ copies mL$^{-1}$. Tukey's pairwise Q showed that the Water 2 sandstone microcosms had statistically significantly higher number of *dsr*B gene copies than all of the Water 1 microcosm samples and the Water 2 no-rock microcosm ($p < 0.02$). The 16S rRNA gene copy mL$^{-1}$ to *dsr*B gene copies mL$^{-1}$ was 17 and 172 16S rRNA genes for each *dsr*B gene copy in the Water 1 and Water 2 mine water, respectively, 472 and 122 in the Water 1 and Water 2 sandstone microcosms, respectively, 66 and 46 in the black shale microcosms and 59 and 25 in the no-rock microcosms, respectively (Figure 5).

## 4. Discussion

In situ, or, in place bioleaching is a technique that is receiving increasing notice as high grade, easily reachable ore resources are growing scarce. This practice includes fracking deeply buried ore formations, which exposes buried rock material to e.g., water, nutrients and microorganisms as well as may introduce blasting residues to the deep biosphere. The newly exposed rock surfaces also provide new surfaces for microorganisms to attach to and may serve as a source of nutrients for the indigenous microbial communities. Both indigenous and introduced microbial communities may affect their living environments and one of the more serious effects is the development of acid rock drainage, ARD. In this study, we aimed to investigate whether the water found in the Rudna mine, which was used as a test site for a pilot in situ copper leaching rock reactor, contained indigenous microbial communities, which could cause ARD and affect the safe decommissioning of the in situ rock reactor. For this purpose, we constructed microcosms containing freshly crushed and washed sandstone or black shale from the Kupferschiefer from the Rudna mine, which we incubated with saline water from two different water collection ponds in the Rudna mine. The pH in the microcosms was followed over time and the number of bacteria and sulphate reducers was examined and the bacterial community composition at the end of incubation period was characterized in order to evaluate the leaching of metals from newly exposed ore surfaces at approximately neutral pH and the risk of ARD formation by indigenous microbial communities.

The diversity of microbial communities in the mine waters and in the microcosm waters was high, with almost 1000 identified genera. There were only minor differences in the community composition between the two original mine water types, despite the almost 4-fold higher salinity in Water 1 compared to the Water 2. The Cu, Zn, Ni and Pb concentrations were also one order of magnitude higher in Water 1 compared to the Water 2. The number of bacteria was one order of magnitude higher in Water 1 than in Water 2, but the incubation with sandstone significantly increased the number of bacteria in the Water 2 microcosms ($p < 0.005$), and although not statistically significantly so, the black shale also increased the number of bacteria in the Water 2. Interestingly, the bacterial diversity in the Water 2 microcosms decreased over the 18 weeks of incubation, as Proteobacteria was the main phylum that benefitted from the conditions (Table 3, Figures 3 and 5). In the Water 1 microcosms, the bacterial numbers increased when incubated together with sandstone and decreased with black shale, compared to the original mine water, although the change was not statistically significant. This indicates that the sandstone may contain substances that induce bacterial growth. In addition, the pH of the microcosm water increased slightly over time in the Water 1 microcosms with sandstone and black shale to reach pH values above 7, whereas in the Water 2 microcosms, the pH slightly decreased. This may also have affected the size of the bacterial community, as the highest numbers of bacterial 16S rRNA gene copies mL$^{-1}$ were detected at pH 7.2–7.5, with the exception of the original Water 2 water and the Water 2 no-rock microcosm. However, a strong pH decrease was not observed, which may be due to the strong buffering capacity of the carbonate rich rock. The difference in sample volumes between the original mine water samples and the microcosm samples, 1000 mL vs. 10 mL, did not appear to markedly affect the detection of bacterial taxa or numbers.

The concentration of Cu in the original mine waters and the microcosm water samples differed greatly, even in the no-rock microcosms. This is most likely due to the use of unfiltered sample water with the HACH kit, as no colloids or precipitates were removed before the measurements. Nevertheless, copper was leached from the rock in both water types, and in both the live and abiotic microcosms. Thus, the leaching of copper was not restricted to only microbial activity, as the Cu concentrations in the abiotic microcosms appeared to be even higher that in the live microcosms (Table 2). This is in accordance with Bomberg et al. [15], where it was shown that Cu was leached from the sandstone both in sterile water and in sterile 10% NaCl solution.

Culturable copper tolerant microorganisms were obtained from some of the microcosms (Table 3). The number of cfu mL$^{-1}$ was very low but indicates nevertheless that the indigenous microbial community has tolerance for even quite high concentration of copper, which may be leached during possible ARD, either due to biological factors or due to newly exposed rock coming into contact with chloride rich mine water. However, it is likely that these microorganisms only tolerate elevated Cu concentrations in their living habitat but may not cause leaching of copper from the rock. Nevertheless, Dziewit et al. [18] found copper resistant/tolerant bacterial strains in the Lubin copper mine, Poland, which tolerated copper concentrations of 2–10 mM. The authors investigated the plasmidome in samples from the Lubin mine and found numerous multi-metal resistance genes situated on plasmids, which may assist the microbial communities to adapt to increased copper concentrations by cells acquiring suitable plasmids.

There was a divergence in the microbial community of one of the Water 1 no rock microcosms, compared to the rest of the Water 1 microcosms (Figure 3). In this microcosm, Water 1 NR B, the bacterial community was similar to those communities detected in the Water 2 microcosms. In addition, the Water 2 microcosm communities had changed markedly from the community in the original mine water over the 18 weeks of incubation. These microcosms, as well as the Water 1 NR B, had in general lower concentrations of Cu than the Water 1 microcosms. Because the microbial communities in both the original mine waters were very similar and the pH did not significantly change over the time of incubation, we suggest that in this case copper may have been the factor that shaped the microbial communities in the microcosms. The mine water used in the microcosms was not filtered and it is possible that the solid substances present in the water may have been unevenly distributed and thus microcosm Water 1 NR B may have had less copper containing solids than its counterpart. The fact that the water used was not filtered may also explain the discrepancy between the concentration of soluble copper in the mine water analyses compared to the copper measured from the microcosm water samples.

Despite the aerobic conditions of the mine water and the aerobic incubation of the microcosm, both mine waters as well as the sandstone and black shale containing and one no-rock Water 1 microcosms showed a surprisingly high content of sulphate reducing bacteria belonging to the phylum Desulfobacterota, contributing with 7.7 to 10.5% of the bacterial 16S rRNA gene sequence reads in these samples. In addition, putative sulphate reducers of the phylum Nitrospirota (genus *Termodesulfovibrionia*), contributed with up to 3.3% of the bacterial 16S rRNA gene sequence pool in these samples. This is also in agreement with the qPCR results showing up to $1.8 \times 10^5$ *dsr*B gene copies mL$^{-1}$ in the original Water 1 (Figure 5). The original Water 2 had only a low content of *dsr*B gene copies, but the number increased in the microcosms over the time of incubation. The 16S rRNA gene to *dsr*B gene ratio in the original mine waters was 17:1 and 172:1 in the Water 1 and Water 2, respectively. The concentration of sulphate was also higher in Water 1, although not markedly so. The sandstone appeared to induce growth of non-sulphate reducing microorganisms more than sulphate reducing microorganisms, as the 16S rRNA gene:*dsr*B gene ratio increased to 472:1 in these microcosms, but not in the Water 2 sandstone microcosms. In contrast, black shale appeared to have similar effect on the 16S rRNA gene:*dsr*B gene ratio in both water types, with 66 or 46 16S rRNA gene copies for every *dsr*B gene copy in Water 1 and Water 2 microcosms, respectively. Some sulphate reducing bacteria have been reported to have different mechanisms to cope with oxygen [22,31,32]. Active sulphate reducers have been detected in the oxic water strata of the Black Sea [33] and Baltic Sea [34] and SRB survive oxygen pulses in anaerobic activated sludge [35].

The bioreactors that would be used for extracting the copper from the in situ rock reactor, contain acidophilic, iron and/or sulphur oxidizing microorganisms that re-oxidize the ferric iron of the lixiviant solution so that it can be re-used in the in situ rock reactor [36]. During operation, acidophilic microorganisms may escape the bioreactor together with the lixiviant and end up in the in situ rock reactor and cause the leaching reaction to continue uncontrolled and thus cause unwanted ARD post operations. It is thus important that the

decommissioning of the in situ rock reactor is performed properly and several studies have shown that specific chemical compounds or chloride-rich water may inhibit the acidophilic iron/sulphur oxidizers permanently (e.g., [14,15]). However, the indigenous microbial communities are likely to change in composition as a result of intrusive mining operations with blasting of rock and exposure of the new rock surfaces to water and the attention of microorganisms, e.g., [37–39]. In our study, no acidification due to the indigenous microbial communities was detected over 18 weeks of immersion of the rock material in Rudna mine water. In order to mimic the situation where new rock is exposed to the surrounding water environment, e.g., in a situation where rock is fractured outside of the immediate effective site of an in situ rock reactor, or when rock is fracked for other mining purposes, the rock material in our study was not acid treated to remove carbonates. As the rock material was not acid treated, as it would be when an in situ rock reactor was constructed in carbonate rich rock, the naturally occurring carbonate in the rock remained to buffer the system and the pH remained above neutral for the whole span of the experiment. This would effectively inhibit any escaped bioreactor bacteria, if they reached untreated rock and the acidic lixiviant was diluted enough. The indigenous microbial communities also did not contain any bacterial taxa known to be acidophilic, and iron and/or sulphur oxidizing (Figure 4). Zhang et al. [16] were able to enrich an acidophilic and thermophilic community dominated by Thermoplasma from Rudna mine Kupferschiefer, which indicates that such microorganisms may exist in the mine and could develop if the pH in the exposed, blasted areas became acidic. However, in our experiment, no known acidophilic, iron/sulphur oxidizing bacterial 16S rRNA gene sequences were found.

## 5. Conclusions

The microbial communities of two chloride-rich mine waters were diverse, but only 46 genera out of 962 identified genera constituted the main bacterial community. The size of the bacterial communities was $3.5 \times 10^5$ and $2.5 \times 10^4$ 16S rRNA gene copies mL$^{-1}$ in Water 1 and Water 2, respectively, and increased by up to 2 orders of magnitude in the sandstone microcosms. Both mine waters had high relative abundance of sulphate reducing bacteria, which decreased in the Water 2 microcosms during the incubation period. However, the number of detected *dsr*B gene copies was high ($1.8 \times 10^4$ mL$^{-1}$) in the original Water 1 and decreased during the incubation period, whereas in Water 2 the number of *dsr*B gene copies was low ($1.5 \times 10^2$ mL$^{-1}$) in the original water and increased during the incubation. No indication of the presence of acidophilic iron oxidizers was found in the mine water or microcosm, but the original Water 1 and part of the Water 2 microcosms, especially the sandstone microcosms, contained Cu tolerant bacteria. The pH in the microcosms did not decrease over time indicating that the chloride-rich mine water and indigenous microorganisms did not acidify the water. In addition, copper was leached from the rock also in the abiotic control microcosms, indicating that the leaching was chemical, not biological.

**Author Contributions:** Conceptualization, M.B., H.M. and P.K.; methodology, M.B. and H.M.; formal analysis, M.B.; investigation, M.B. and H.M.; resources, P.K.; writing—original draft preparation, M.B.; writing—review and editing, M.B., H.M. and P.K.; project administration, P.K.; funding acquisition, P.K. All authors have read and agreed to the published version of the manuscript.

**Funding:** This research was funded by European Union's Horizon 2020 Research and Innovation program under Grant Agreement # 642456, BIOMOre project.

**Data Availability Statement:** The sequence data in this paper has been deposited in the European Nucleotides Archive (ENA, https://www.ebi.ac.uk/ena/, accessed on 14 March 2022) under Study number PRJEB47295.

**Acknowledgments:** Mirva Pyrhönen (VTT), Veera Partanen (VTT) and Tuula Kuurila (VTT) are thanked for excellent laboratory work. The Rudna Mine is thanked for providing sample water and rock material.

**Conflicts of Interest:** The authors declare no conflict of interest.

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
