# Peer review of "The Diverse Indigenous Bacterial Community in the Rudna Mine Does Not Cause Dissolution of Copper from Kupferschiefer in Oxic Conditions"

_minerals, doi:10.3390/min12030366_

Round 1
Reviewer 1 Report
Thew manuscript investigates the indigeneous bacterial community of Rudna mine samples (waters from Rudna mine and sandstone and black shale). Work includes microcosm experiments, cultivation experiments and DNA analyses. The manuscript is well structured and wititen, research design is comperehensive.
Author Response
Thank you!
Reviewer 2 Report
Bomberg et al present a well-written manuscript. It is, of course, very difficult to prove a negative, and so it is important to be circumspect in the conclusions.
While I believe that the authors are correct in their conclusions (on the balance of probability), I still have some concerns/comments I would like to see addressed.
Firstly, the microcosms used appear to be microaerobic at best, and I would imagine very nutrient limited. Do the authors believe that 18 weeks is long enough to evaluate the establishment of potentially mineral-oxidising communities?
Why was control experiments not done under more favourable conditions, to show whether the capacity exists in the community – for example using aearated, lower pH systems (in other words try to enrich mineral-oxidisers from the water). Negative results here would be more definitive in suggesting that the mineral-oxidising capacity does not exist in the community (mine water). Positive results here, would reinforce the importance of controlling the oxygen/pH.
Along the same line, in the in situ leaching work, the buffering capacity of the rock was removed with sulfuric acid. In the microcosms, it wasn’t, and the system remained buffered. Furthermore, in the non-rock samples (water 1 and water 2) significant pH decrease was observed.
What is the source of this acidity?
Why was acid-treated rock not used to better simulate the actual, post-leaching, system? If pH drops, do mineral-oxidising organisms become established? For me this is quite an important short-coming.
The study demonstrates that Cu is leached from the rock, even a neutral pH. This needs to be commented on, as this drainage is therefore contaminated, and would still need to be treated. It’s just not acidic…
I’m not sure of the added value of Fig 3. What are you trying to show? One can go further, even very simplistically, using envfit to see if community structure correlates with specific environmental factors and even something like ANOSIM to evaluate the significance of any clustering observed. But again, what are you trying to analyse?
It seems important that no known mineral-oxidising organisms were detected in the water samples, though again, difficult to prove an absolute absence.
In the real system, there would be continual percolation of the water through the rock, whereas in this study you have just use replicate 100 mL samples. Is this representative?
The abundance of SRB is interesting and implies that the waters are naturally anoxic? Is this the case? It wasn’t clear in the description. Can you speculate on the role these organisms may play, and what their energy sources may be?
Overall, I think the study should be published, as it is an interesting piece of work with implications beyond the BioMORe project. However, I would like to see the replies to my queries above.
Author Response
Reviewer 2
Comment: Bomberg et al present a well-written manuscript. It is, of course, very difficult to prove a negative, and so it is important to be circumspect in the conclusions.
While I believe that the authors are correct in their conclusions (on the balance of probability), I still have some concerns/comments I would like to see addressed.
Response: We would like to thank the Reviewer for insightful comments on out manuscript. We have revised the manuscript according to the Reviewer’s suggestions and comments to the best of our ability.
Comment: Firstly, the microcosms used appear to be microaerobic at best, and I would imagine very nutrient limited. Do the authors believe that 18 weeks is long enough to evaluate the establishment of potentially mineral-oxidising communities?
Response: One of the major focus points of the Biomore project was the decommissioning of introduced iron and sulfur oxidizing acidophilic microorganisms that were used in an on-site ferric iron generating bioreactor through which the lixiviant was led to and from the in situ rock reactor in the copper ore layer in the Rudna mine. Due to the high content of carbonates in the rock the in situ rock reactor was acid washed before the actual leaching of copper could occur, making at least the pH conditions optimal for escaped mineral oxidizing acidophiles to survive after end of operations. In our previous laboratory experiment (Bomberg, M., Miettinen, H., Wahlström, M., Kaartinen, T., Ahoranta, S., Lakaniemi, A. M., & Kinnunen, P. (2018). Post operation inactivation of acidophilic bioleaching microorganisms using natural chloride-rich mine water. Hydrometallurgy, 180, 236-245.) with an experimental setup similar to this one but with acid treated rock material, we saw that thee introduced acidophiles did not survive after treatment of the microcosms with the near neutral pH Cl-rich mine water also used in this study (Water 1).
However, the question arose that what happens in areas in the rock that fracture due to the fracking activity taking place when the rock reactor is constructed but to where the acid treatment and leaching solutions do not reach but natural mine water collects? Our present experiment tests the possibility of spontaneous leaching in this situation, which would most likely be restricted in oxygen although not absolutely anoxic, and where the main microorganisms would be those indigenous to the environment and that could benefit from freshly exposed rock and mineral surfaces in an otherwise nutrient limited environment.
It is possible that 18 weeks may not be long enough to see the establishment of mineral oxidizing communities, but this was how long we could run the experiment and over this period of time it didn’t happen.
Comment: Why was control experiments not done under more favourable conditions, to show whether the capacity exists in the community – for example using aearated, lower pH systems (in other words try to enrich mineral-oxidisers from the water). Negative results here would be more definitive in suggesting that the mineral-oxidising capacity does not exist in the community (mine water). Positive results here, would reinforce the importance of controlling the oxygen/pH.
Response: We actually set up an enrichment experiment using acidic mineral medium with ferrous iron in order to test for acidic iron oxidizing activity in the microcosms at the end of the incubation period. The result was negative, no iron oxidizing acidophiles or iron oxidizing activity emerged. This has been added to the manuscript (L154 – 169, L290 – L293).
Comment: Along the same line, in the in situ leaching work, the buffering capacity of the rock was removed with sulfuric acid. In the microcosms, it wasn’t, and the system remained buffered. Furthermore, in the non-rock samples (water 1 and water 2) significant pH decrease was observed.
Response: As explained above, with this experiment we wanted to test the situation outside of the in situ bioleaching reactor to see what could happen in places where the rock is freshly fractured and subjected to the natural water and microbes in the mine.
Comment: What is the source of this acidity?
Response: Unfortunately, this is not known. It may be that the lack of buffering from the rock was enough to cause this pH drop.
Comment: Why was acid-treated rock not used to better simulate the actual, post-leaching, system? If pH drops, do mineral-oxidising organisms become established? For me this is quite an important short-coming.
Response: As explained above, this was not the focus of this experiment. We have previously shown (Bomberg et al., 2018) that with acid treated rock and the pH around 2.5 no iron oxidizing activity was observed in microcosms containing Water 1. At this point the rock had disintegrated to silt due to the acid treatment. In the present experiment the rock still contained all potential nutrients the microorganicms could benefit from.
Comment: The study demonstrates that Cu is leached from the rock, even a neutral pH. This needs to be commented on, as this drainage is therefore contaminated, and would still need to be treated. It’s just not acidic…
Response: Yes, this is the situation. The chloride in the water causes this leaching and has been discussed in Bomberg et al., 2018.
Comment: I’m not sure of the added value of Fig 3. What are you trying to show? One can go further, even very simplistically, using envfit to see if community structure correlates with specific environmental factors and even something like ANOSIM to evaluate the significance of any clustering observed. But again, what are you trying to analyse?
Response: We are showing the similarity/dissimilarity of the bacterial community composition in the original water compared to the community developed in the microcosms during incubation. As seen from the plot the bacterial communities in water 2 changed over time compared to the original water and formed a tight cluster of their own on the plot. This development was apparently not caused by contact with newly exposed rock, since also the no-rock communities fell into the same group. However, the water 1 sandstone microcosm community remained similar to the original water 1 community, whereas the communities in the water 1 black shale and no rock microcosms developed in a different direction.
Unfortunately, the Cu measurements of the original water samples and the microcosm water samples were not done in a similar way. We got the water chemistry data from the mine without proper knowledge of how the measurements were done. Most likely with ICP-EOS using water samples filtered with 0.45 µm pore-size filters. However, we did the Cu measurement with the Hach kit on diluted unfiltered samples and the Cu concentration differed by 100x between the measurements we got from the mine and our no-rock microcosm water. Thus, any statistical comparison of the Cu content between the original water and microcosm would be meaningless. The pH was tested, but there was no statistically significant effects of pH on the community composition.
If deemed unnecessary the Figure can be removed.
Comment: It seems important that no known mineral-oxidising organisms were detected in the water samples, though again, difficult to prove an absolute absence.
Response: This is true. We are limited by our experimental setup, but with these tests and methods we couldn’t detect any presence of mineral-oxidizing microbial activity.
Comment: In the real system, there would be continual percolation of the water through the rock, whereas in this study you have just use replicate 100 mL samples. Is this representative?
Response: No, but this is a start. A real system would not have been possible at this point because we were also limited by the amounts of test material we could get.
Comment: The abundance of SRB is interesting and implies that the waters are naturally anoxic? Is this the case? It wasn’t clear in the description. Can you speculate on the role these organisms may play, and what their energy sources may be?
Response: A clarifying sentence has been added to the M&M section 2.1 where the constant exposure of the ponds to the ambient atmosphere is clarified (L123 – L124).
Yes, the SRB were a surprise. Interestingly these bacteria were common in the original mine waters, which were ponds where water collected in the mine and as such subjected to the ambient oxygen levels. The oxygenation level of the waters is not known, but likely not restricted. Sulphare was abundantly available. As far as one can speculate about energy sources, there were quite abundantly volatile substances available, which could be for example VOCs, which may originate from the bituminous material in the rock (see clarification in the introduction). In addition, the rock material contained high amounts of carbonates and the rocks could also have contained residual recalcitrant carbon compounds. Several species of the groups detected in these samples have the capacity for autotrophic growth, such as some Desulfobacter and Desulfosarcina spp. However, as the SRB are not the main or sole focus of this experiment and as the energy and carbon sources used by the other bacteria have not been addressed speculations about the SRB energy sources may be a bit excessive?
Comment: Overall, I think the study should be published, as it is an interesting piece of work with implications beyond the BioMORe project. However, I would like to see the replies to my queries above.
Reviewer 3 Report
“The diverse indigenous bacterial community in the Rudna mine does not induce ARD in oxic conditions”
Title: please do not use acronyms in the title
line 70: briefly describe what Kupferschiefer is. Also please provide some characteristics of the sandstone and black shale in order to better understand the natural system and the microcosms assays
line 121: what about aeration of the microcosms?
line 133: it is well known that sulphide mining environments present low quantities of organic matter, therefore, heterotrophic microorganisms are not quite abundant, and generally they are not responsible for the main processes, such as acid rock drainage; thus, can the authors explain why did they decided to culture heterotrophic cupper resistant microorganisms?
lines 249-253: the overestimation in Cu concentration in the NR microcosms compared to the Cu concentrations measured some other way (it should be al least mentioned how) is of about 2 orders of magnitude, which deserves further discussion. Do the authors measure the Cu concentration of the waters with the HACH method?
line 254: it is not clear why is it important to determine the number of Cu tolerant culturable, heterotrophic bacteria. If there is a reason it should have been explained in the introduction.
line 272: it is not clear why is it relevant to particularly detect sulphate reducing bacteria
Discussion 1st paragraph: the description and aim of the work is clearly presented, and in spite of not being original, in this particular case it surprising, or at least not explained, why the authors look for acid rock drainage in the microcosms using waters of neutral pH and for a relatively short period of time. Even more, the high throughput sequencing of all the samples show that there are no microbial species with the capacity to generate acid rock drainage among the over 1% most abundant species. It is also not clear why in the context of the manuscript the authors focus the attention on sulphate reducing bacteria.
In the last paragraph of the discussion the authors comment some of the issues presented above, however in my opinion, they are not conclusions extracted after deep analysis of the result and bibliographic evidence, but rather obvious facts derived from basic information of the assays (pH of the waters, carbonate content of the mineral, composition of the microbial communities).
The aim of the paper, even though not original, it is interesting from environmental and biotechnological points of view; however, in this case the lack of ARD can be explained by the neutral pH of the waters, the high concentration of carbonates in the minerals and the lack of acidophilic iron and/or sulphur oxidising species in the original samples (as in fact the authors did in the last paragraph of the discussion) with no real need of the microcosms assays.
In my opinion the manuscript needs to be rewritten considering a modification of its aim, and it would surely benefit from a combination of the results and discussion section
Author Response
Reviewer 3
“The diverse indigenous bacterial community in the Rudna mine does not induce ARD in oxic conditions”
Comment: Title: please do not use acronyms in the title
Response: We would like to thank the reviewer for insightful comments. The title of the manuscript has been changed to focus on Cu dissolution, not ARD.
Comment: line 70: briefly describe what Kupferschiefer is. Also please provide some characteristics of the sandstone and black shale in order to better understand the natural system and the microcosms assays
Response: Some clarification has been added to the Introduction (L71 – L76) and the composition of the sandstone and black shale has been added to Table 1.
Comment: line 121: what about aeration of the microcosms?
Response: The microcosms were incubated with the caps loosely screwed on so that some ventilation could occur, but not so much as to cause much evaporation. This is now clarified in the text (L132-133). The bottles were also opened on occasion for taking samples for pH and Cu measurements allowing for more thorough ventilation during the sampling.
The microcosms were not actively aerated because we wanted to mimic the situation caused by fracking inside a rock reactor that would not be constantly exposed to oxygen but also not absolutely anoxic.
Comment: line 133: it is well known that sulphide mining environments present low quantities of organic matter, therefore, heterotrophic microorganisms are not quite abundant, and generally they are not responsible for the main processes, such as acid rock drainage; thus, can the authors explain why did they decided to culture heterotrophic copper resistant microorganisms?
Response: We wanted to obtain more information about the microbial communities in the Cu rich mine water in general test for the possibility of copper resistance using a medium that is generally used for culturing environmental microorganisms. As shown by the added data in Table 1, the rocks themselves contained quite a lot of organic compounds and Cu resistant bacteria have previously also been shown to grow on organic carbon rich media, e.g.
https://sfamjournals.onlinelibrary.wiley.com/doi/abs/10.1111/j.1365-2672.2009.04261.x
https://www.frontiersin.org/articles/10.3389/fmicb.2015.00152/full
In addition, we tested to enrich acidophilic microorganisms on mineral media (added to the manuscript, L154 – 169, L290 – L293) but did not see any response.
Comment: lines 249-253: the overestimation in Cu concentration in the NR microcosms compared to the Cu concentrations measured some other way (it should be al least mentioned how) is of about 2 orders of magnitude, which deserves further discussion. Do the authors measure the Cu concentration of the waters with the HACH method?
Response: As explained in the Table 1 legend, the physicochemical measurements are from Szubert et al. and measured by the mining company. Unfortunately, the methodology is not available to us. The method was most likely with ICP-EOS using water samples filtered with 0.45 µm pore-size filters, but as we are not absolutely certain we cannot comment on this.
We measured the Cu concentration of the water samples from the microcosm using the HACH method and used unfiltered sample for the measurements. It is possible that this affected the results. It is mentioned in the Materials and Methods that the samples were unfiltered. In addition, a comment was added to the results and the discussion. Even with this discrepancy it is clear that the more chloride rich water and at the end of the incubation, the sandstone microcosm waters were visibly green.
Comment: line 254: it is not clear why is it important to determine the number of Cu tolerant culturable, heterotrophic bacteria. If there is a reason it should have been explained in the introduction.
Response: We wanted to have as much relevant information as possible about the microorganisms inhabiting the saline mine waters in the Rudna mine and as Cu is generally considered toxic to microorganisms, the fact that there was a rich microbial community present in these waters is interesting and these bacteria may have some specialized ways to cope with elevated Cu concentration. Similar tests have been done in the Lubin mine and the authors used organic carbon rich media because there is a surprisingly high content of organic carbon in the Kupferschiefer that originates from the bituminous residues of the rock.
https://sfamjournals.onlinelibrary.wiley.com/doi/abs/10.1111/j.1365-2672.2009.04261.x
https://www.frontiersin.org/articles/10.3389/fmicb.2015.00152/full
A paragraph has been added to the Introduction (L102 – 106).
Comment: line 272: it is not clear why is it relevant to particularly detect sulphate reducing bacteria
Response: The sulphate reducers were a surprise. After obtaining the sequence results that showed the presence of these bacteria in the original water as well as in the microcosm water we wanted to know the size of the SRB community compared to the general bacterial community.
Comment: Discussion 1st paragraph: the description and aim of the work is clearly presented, and in spite of not being original, in this particular case it surprising, or at least not explained, why the authors look for acid rock drainage in the microcosms using waters of neutral pH and for a relatively short period of time. Even more, the high throughput sequencing of all the samples show that there are no microbial species with the capacity to generate acid rock drainage among the over 1% most abundant species. It is also not clear why in the context of the manuscript the authors focus the attention on sulphate reducing bacteria.
Response: We previously showed that acidophilic, iron oxidizing microorganisms originating from a ferric iron generating bioreactor, used to produce ferric lixiviant for in situ bioleaching of the same ore as used in this study, were effectively inhibited when immerged in chloride-rich water, i.e. water 1 used in this study (Bomberg, M., Miettinen, H., Wahlström, M., Kaartinen, T., Ahoranta, S., Lakaniemi, A. M., & Kinnunen, P. (2018). Post operation inactivation of acidophilic bioleaching microorganisms using natural chloride-rich mine water. Hydrometallurgy, 180, 236-245.). In the previous experiment the buffering capacity of the rock was removed with sulfuric acid and the conditions in the microcosms were maintained at low pH over the bioleaching phase after which the process was terminated by adding water 1 to the microcosms. The pH didn’t rise in the experimental microcosms, but the iron oxidizing activity ceased. However, using chloride free water the iron oxidizing activity continued, which is not surprising. With the present experiment, we wanted to examine whether the indigenous microorganisms could affect the dissolution of Cu or cause acidification in a situation where the rock has been fractured for in situ bioleaching, but to where the actual bioleaching or post operation decommissioning solutions could not reach and the naturally occurring microorganisms in the percolating water could benefit from the newly exposed rock surfaces. They did, but they did not overpower the buffering capacity of the rock at least not over the 18 weeks of the experiment. However, the chloride of the water caused leaching of Cu from the rock, which may be a bigger problem than the microorganisms.
Comment: In the last paragraph of the discussion the authors comment some of the issues presented above, however in my opinion, they are not conclusions extracted after deep analysis of the result and bibliographic evidence, but rather obvious facts derived from basic information of the assays (pH of the waters, carbonate content of the mineral, composition of the microbial communities).
Response: These are the conclusions and summarizations of the work, based on the outcome of the study and not on speculations.
Comment: The aim of the paper, even though not original, it is interesting from environmental and biotechnological points of view; however, in this case the lack of ARD can be explained by the neutral pH of the waters, the high concentration of carbonates in the minerals and the lack of acidophilic iron and/or sulphur oxidising species in the original samples (as in fact the authors did in the last paragraph of the discussion) with no real need of the microcosms assays.
Response: The title of the manuscript was changed to focus on Cu dissolution instead of ARD.
Comment: In my opinion the manuscript needs to be rewritten considering a modification of its aim, and it would surely benefit from a combination of the results and discussion section
Response: The results and discussion will be left separate for clarity.
Round 2
Reviewer 3 Report
Please either use f or ph to write sulphur and derived words. Chek all through the text
Abstract, line 32-35: the fact that Cu was leached from the negative control does not automatically discards leaching caused by bacteria. Please rewrite the sentence including some quantitative information
Line 138-140: please give a brief explanation why there was only one replica of the abiotic microcosm with water 1.
Check that various sections are numbered 3.3
“Abundance of microorganisms” section: the quantification of the dsrB gene is out of context in this part of the manuscript. There is no mention to SRB in the introduction so the reader inevitably wonders why this particular group is worth of attention. If, as the authors mentioned in the review 1 response letter, the high abundance of SRB was a surprise of the 16S rRNA sequencing, this whole section should appear after the Sequence data and Microbial community sections. Otherwise, it is not clear why the dsrB was quantified.
In fact, there is no special mention to the SRB in the section that describes the microbial community and their relative abundance in Figure 5 does not seem particularly high.
Line 344: which concentration are they referring to?
Line 505: Please explain why the rock material was not acid treated, if this would be done when an in situ rock reactor had been constructed in carbonate rich rock. Such treatment would probably modify the pH behaviour and therefore modify the microbial community and the capacity of generating acid rock drainage
Author Response
Please either use f or ph to write sulphur and derived words. Chek all through the text
Response; Checked & corrected to ph
Abstract, line 32-35: the fact that Cu was leached from the negative control does not automatically discards leaching caused by bacteria. Please rewrite the sentence including some quantitative information
Response; The sentence was revised accoding to suggestion.
Line 138-140: please give a brief explanation why there was only one replica of the abiotic microcosm with water 1.
Response; Explanation added. Unfortunately, we did not get more sample water.
Check that various sections are numbered 3.3
Response; Corrected
“Abundance of microorganisms” section: the quantification of the dsrB gene is out of context in this part of the manuscript. There is no mention to SRB in the introduction so the reader inevitably wonders why this particular group is worth of attention. If, as the authors mentioned in the review 1 response letter, the high abundance of SRB was a surprise of the 16S rRNA sequencing, this whole section should appear after the Sequence data and Microbial community sections. Otherwise, it is not clear why the dsrB was quantified.
In fact, there is no special mention to the SRB in the section that describes the microbial community and their relative abundance in Figure 5 does not seem particularly high.
Response; The combined relative abundance of the sequences belonging to Desulfobacterota and Thermodesulfovibriona comprised represented more than 10% of the total bacterial community in the original water samples and the rock microcosms with Water 1. We consider this a surprisingly high portion of the bacterial community in this environment. A paragraph was added to the text to eexplain this. We agree with the reviewer that the order of the sections should change. The “Abundance of microorrganisms” section has been moved to the end of the results section and the numbering of the figures has been changed accordingly.
Line 344: which concentration are they referring to?
Response; This text was left in the manuscript by mistake and has been removed.
Line 505: Please explain why the rock material was not acid treated, if this would be done when an in situ rock reactor had been constructed in carbonate rich rock. Such treatment would probably modify the pH behaviour and therefore modify the microbial community and the capacity of generating acid rock drainage
Response; An explanation has been added. We have tested the same with acid treated rock and Water 1 (https://www.sciencedirect.com/science/article/abs/pii/S0304386X18300768).
Round 3
Reviewer 3 Report
The added phrases must be polished to be integrated to the rest of the manuscript. In the actual version they look like pasted without been read twice